# Cost-effectiveness of Hepatitis C virus self-testing in four settings

**Josephine G. Walker**[1]*, **Elena Ivanova**[2], **Muhammad S. Jamil**[3], **Jason J. Ong**[4,5],
**Philippa Easterbrook**[3], **Emmanuel Fajardo**[2,3], **Cheryl Case Johnson**[3], **Niklas Luhmann**[3],
**Fern Terris-Prestholt**[6], **Peter Vickerman**[1], **Sonjelle Shilton**[2]

1 Population Health Sciences, Bristol Medical School, University of Bristol, Bristol, United Kingdom, 2 FIND, The Global Alliance for Diagnostics, Geneva, Switzerland, 3 Global HIV, Hepatitis and STI Programmes, World Health Organization, Geneva, Switzerland, 4 London School of Hygiene & Tropical Medicine, London, United Kingdom, 5 Central Clinical School, Monash University, Melbourne, Australia, 6 Joint UN programme on HIV and AIDS (UNAIDS), Geneva, Switzerland

* Josephine.Walker@bristol.ac.uk

**Data Availability Statement:** The model and data are available at https://github.com/jogwalker/HCVST.

**Funding:** Unitaid and the Netherlands Government funded this study as part of the HEAD-Start project

## Abstract

Globally, there are approximately 58 million people with chronic hepatitis C virus infection (HCV) but only 20% have been diagnosed. HCV self-testing (HCVST) could reach those who have never been tested and increase uptake of HCV testing services. We compared cost per HCV viraemic diagnosis or cure for HCVST versus facility-based HCV testing services. We used a decision analysis model with a one-year time horizon to examine the key drivers of economic cost per diagnosis or cure following the introduction of HCVST in China (men who have sex with men), Georgia (men 40–49 years), Viet Nam (people who inject drugs, PWID), and Kenya (PWID). HCV antibody (HCVAb) prevalence ranged from 1%-60% across settings. Model parameters in each setting were informed by HCV testing and treatment programmes, HIV self-testing programmes, and expert opinion. In the base case, we assume a reactive HCVST is followed by a facility-based rapid diagnostic test (RDT) and then nucleic acid testing (NAT). We assumed oral-fluid HCVST costs of $5.63/unit ($0.87-$21.43 for facility-based RDT), 62% increase in testing following HCVST introduction, 65% linkage following HCVST, and 10% replacement of facility-based testing with HCVST based on HIV studies. Parameters were varied in sensitivity analysis. Cost per HCV viraemic diagnosis without HCVST ranged from $35 2019 US dollars (Viet Nam) to $361 (Kenya). With HCVST, diagnosis increased resulting in incremental cost per diagnosis of $104 in Viet Nam, $163 in Georgia, $587 in Kenya, and $2,647 in China. Differences were driven by HCVAb prevalence. Switching to blood-based HCVST ($2.25/test), increasing uptake of HCVST and linkage to facility-based care and NAT testing, or proceeding directly to NAT testing following HCVST, reduced the cost per diagnosis. The baseline incremental cost per cure was lowest in Georgia ($1,418), similar in Viet Nam ($2,033), and Kenya ($2,566), and highest in China ($4,956). HCVST increased the number of people tested, diagnosed, and cured, but at higher cost. Introducing HCVST is more cost-effective in populations with high prevalence.

led by EI and SS at FIND; with support from the
Unitaid-WHO HIV and Co-Infections/Co-Morbidities
Enabler Grant (HIV&COIMS) led by CJ, MJ, NL, PE
at WHO. PV also acknowledges support from the
NIHR Health Protection Research Unit in
Behavioural Science and Evaluation at the
University of Bristol. The funders had no role in
study design, data collection and analysis, decision
to publish, or preparation of the manuscript.

**Competing interests:** EI and SS are current
employees of FIND, EF is a former employee of
FIND. JGW and PV have received unrestricted
research funding from Gilead Sciences unrelated to
this research. All authors declare no other conflicts
of interest. The views expressed in this manuscript
are those of the authors and do not necessarily
represent the official position, decisions, policy or
views of the WHO or UNAIDS. There are no
patents, products in development or marketed
products associated with this research to declare.
This does not alter our adherence to PLOS ONE
policies on sharing data and materials.

## Introduction

In 2016 the World Health Organization (WHO) launched the Global Health Sector Strategy for Viral Hepatitis with the goal to eliminate viral hepatitis B and C as a public health problem by 2030 [1]. To achieve these elimination targets, the strategy outlined the need to diagnose 90% of infected persons and treat 80% of those diagnosed, alongside scaling up prevention interventions. In 2019, there were approximately 58 million people with chronic HCV globally, of which only 21% had been diagnosed [2].

Reaching individuals not yet tested and aware of their HCV status, including those who are hesitant or unable to access facility-based services, is critical for achieving elimination. Self-testing (a process by which an individual collects his or her own specimen, performs a rapid diagnostic test, and interprets the result) is an additional testing approach to reach those who have never been tested and increase access and uptake of HCV testing services more broadly. During the COVID-19 pandemic in 2020, healthcare systems have been strained, and health-care resources diverted to tackle the pandemic, resulting in reported disruptions to hepatitis diagnosis and treatment programmes in 43% of countries [2,3]. Self-testing for HCV (HCVST) may help sustain or increase HCV testing rates despite these challenges, with appli-cation to both the general population and key populations, such as people who inject drugs (PWID) and men who have sex with men (MSM).

The WHO-recommended diagnostic pathway of testing someone for HCV starts with a sin-gle serological HCV test. Those who have a positive (reactive) result are then tested using HCV RNA nucleic acid testing (NAT), or core antigen, to confirm if viraemic HCV infection is present [4,5]. While HCVST could replace initial provider-administered serology testing, facility-based confirmation of viraemic infection by NAT or core antigen test will still be necessary.

HCVST uses rapid diagnostic tests (RDT), which detect antibodies in fingerprick/capillary whole blood and/or oral fluid. With evidence emerging and quality-assured HCVST products coming to market in the near future, primarily with those adapting professional use tests for self-testing, it is important to consider the potential impact and affordability of introducing HCVST alongside existing services. In 2020, the World Health Organization began developing normative guidance on HCVST introduction [6]. A systematic review conducted as part of the guidance found no previous studies on the cost or cost-effectiveness of using HCVST [7]. To support the development of these guidelines, we conducted a cost-effectiveness analysis on the introduction of HCVST, with self-testing to be performed by an individual with or without direct support. This analysis projects the short-term costs and outcomes of introducing HCVST alongside standard diagnostic pathways, in four low and middle-income settings, including amongst key populations.

## Methods

We developed a decision tree model representing the path from HCV testing to diagnosis, treatment and cure (Fig 1). We modelled four scenarios among different countries and popula-tions with varying HCV Antibody (HCVAb) prevalence: PWID (Viet Nam and Kenya), MSM (China), and men 40–49 years (Georgia), where the national approach to HCV testing and treatment varies (S1 Text). The costs of distributing self-tests and standard of care testing and treatment, and the care cascade differ for each setting based on local data (Table A in S1 Text). Here we report on the cost per diagnosis of viraemic infection and cost per cure, of HCVST compared to standard facility-based HCV testing pathways alone, and do not model long term outcomes such as infections or disability adjusted life years averted. Parameter estimates were gathered from real-world examples using literature on HCV testing and treatment

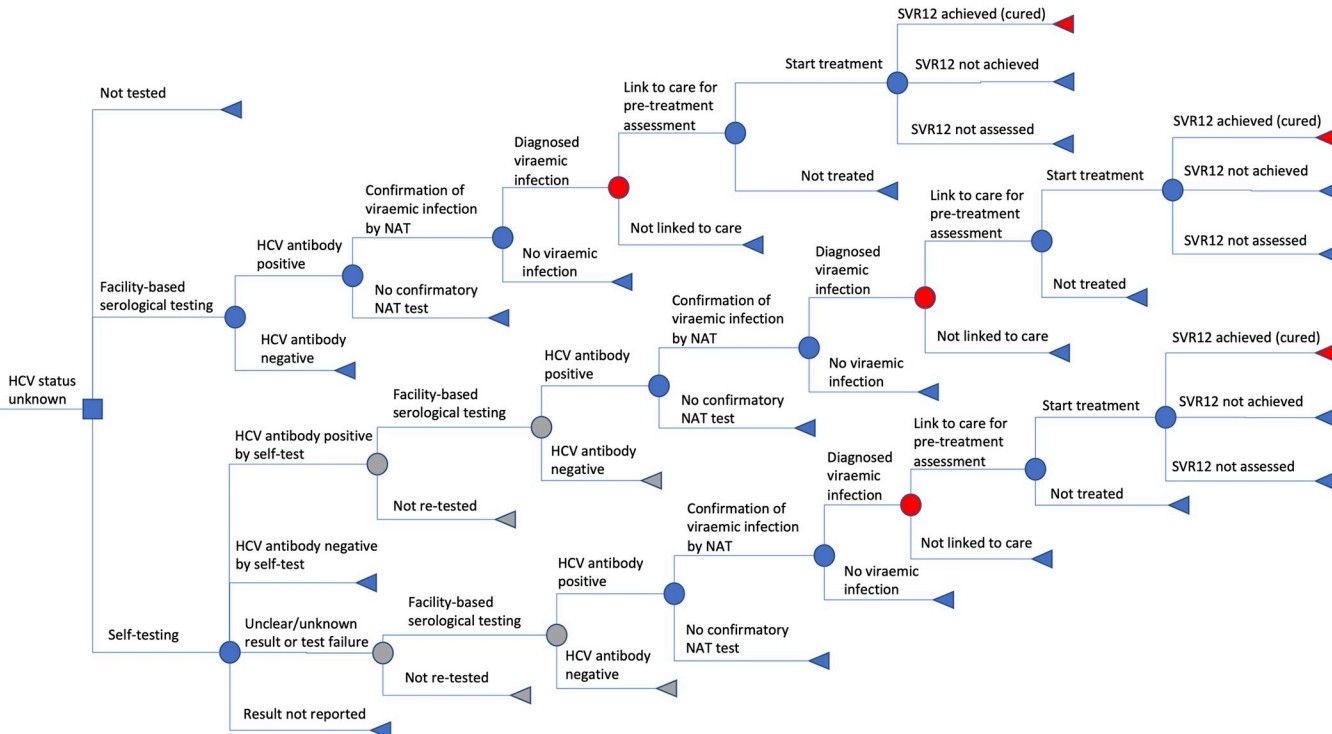

**Fig 1. Decision tree diagram representing the pathways analysed for the introduction of self-testing, including the cascade of care after receiving a positive HCV antibody test through to treatment and cure.** Red nodes represent the two outcomes of interest examined in this study: Diagnosed with viraemic infection, and cure. Grey nodes are removed in the direct to NAT scenario which is examined in sensitivity analysis. Probabilities and costs used in the model are presented in Table 1 and Tables A and B in S1 Text. Abbreviations: HCV = Hepatitis C virus; HCVAb = HCV antibody; NAT = nucleic acid test; SVR12 = sustained virologic response (cure).

programmes and HIV self-testing programmes in each setting, expert opinion, and manufacturers (self-test unit costs). We incorporated data on HCVST usage from recent HCVST feasibility and acceptability trials in the study settings [8].

We explore alternative models and pathways for using HCVST, including whether HCVST is offered alongside HIVST (Kenya and China), whether confirmatory NAT testing for viraemic infection is done immediately on samples from those with a reactive facility-based serologic test (reflex testing; Kenya and Viet Nam), and whether self-tests are peer-guided (Kenya and Vietnam). We assume a positive self-test leads to the standard pathway in which patients arriving at a facility will receive facility-based serologic testing ("repeat serologic testing") prior to confirmatory NAT testing; however we explore in sensitivity analysis a scenario in which positive self-tests receive NAT tests directly. Although core antigen testing is a WHO-recommended option for confirmation of viraemic infection, in this study, we only evaluate pathways using NAT testing as it is the main confirmation method for viraemic infection used in all the case study settings.

## Model structure

The decision tree model (Fig 1) is based on a cross-sectional evaluation of the proportion of the population of interest who do not know their status, using testing and linkage to treatment rates expected within a one year time horizon. The population examined differs by case study setting. The model represents three pathways for the study population: 1) standard of care testing, in which people receive facility-based testing and do not get self-tests; 2) no testing, for the

proportion of the population who remain untested by any method; and 3) self-testing, which provides testing for a subset of the group who otherwise would not access testing, as well as replacing some of the standard of care tests with self-tests.

Those in the no testing pathway are assumed not to be diagnosed or access HCV care in the modelled year. Standard of care testing consists of facility-based anti-HCV antibody testing, with anti-HCV positive individuals receiving confirmatory HCV RNA nucleic acid testing (NAT). Those with confirmed infection are referred for pre-treatment clinical assessment, treated, and then evaluated for a sustained virological response using NAT (typically 12 weeks after end of treatment). Self-testing follows a similar pattern, but self-test results are separated into four possible outcomes depending on the modelled scenario (Fig 1): 1) self-test result conducted but result not reported, 2) positive self-test result with retesting using standard of care anti-HCV antibody testing (repeat serologic testing scenario), 4) negative self-test, which is assumed not to lead to follow up testing, or 5) an invalid test result, such as if the result is not readable by the individual, in which case they would receive repeat serology testing. Details of how parameters and costs are incorporated within the model are given in Tables B and C in S1 Text.

## Analysis

For each setting, we compared the introduction of HCVST in terms of numbers of individuals diagnosed with viraemic infection or cured, total cost, cost per diagnosis, and cost per cure, to a counterfactual standard of care scenario in which no self-testing occurs. Cost per diagnosis, cost per cure and the incremental cost-effectiveness ratio (ICER) were calculated. The ICER divides the difference in total cost between the HCVST and the counterfactual scenario (incremental cost) by the difference in the number of people diagnosed/cured between the HCVST scenario and the no self-testing scenario (incremental effect). This provides a measure of the extra cost per extra person diagnosed or cured with the introduction of HCVST.

**Base case assumptions.** We use the repeat serologic testing pathway as the base case, which assumes that individuals with a reactive self-test presenting to a healthcare facility (which could include "community" testing sites such as harm reduction centres) are tested by the standard of care pathway starting from a facility-based serologic test. This facility-based serologic test is assumed to be the SD Bioline HCV rapid test (Abbott Diagnostics, IL, USA) which has an overall sensitivity of 95% (95% CI 93–96%) and specificity of 100% (95% CI 99–100%) [24].

In the base case HCVST analysis for each setting, we present the cost-effectiveness of using oral-fluid based self-tests (OraQuick HCV Rapid Antibody Test, OraSure Technologies, PA, USA), as these were evaluated in the HCVST usability study [8]. The sensitivity and specificity of the OraQuick test on an oral sample are reported to be 98% (95% CI 97%-99%) and 100% (95% CI 90%-100%), respectively [25]. We adjust the sensitivity and specificity to account for misinterpretation of results during self-testing, as observed by inter-reader agreement in the usability studies in each setting (88–98%, Table 1) [8]. In addition, we make the assumption that 3% of self-tests are used incorrectly so that no result can be reported, but which still lead to the individual linking to facility-based testing (invalid test result). The cost of the oral-fluid based HCVST was estimated to be $4.50 plus 25% overheads to account for human resources and infrastructure ($5.63 total), based on the authors' expert opinion, drawing on experience from HIVST and knowledge of current HCV diagnostic test pricing.

The uptake of self-testing and linkage to care parameters were determined based on randomised controlled trials of HIV self-testing, with 65% of reactive or invalid self-tests linking to facility-based testing, and self-testing leading to a 62% increase in the number of people tested

**Table 1. Assumptions and parameters used in the analysis that vary by country.**

| | Kenya PWID | Georgia men 40–49 | Vietnam PWID | China MSM | Source* |
|---|---|---|---|---|---|
| *Transition parameters* | | | | | |
| Size of study population | 13,450 | 234,200 | 5,000 | 17,000 | [9–12] |
| HCV Antibody prevalence | 13% (11–15%) | 23% (18–29%) | 66% (46–87%) | 1% (0.6–1.5%) | [13–17] |
| Undiagnosed with HCV | 71% | 76% | 70% | 80% | [14,18]* |
| Chronic hepatitis C Prevalence among those Ab+ | 77% | 80% | 84% | 75% | [17–19]* |
| Standard of care test uptake among unknown status per year | 50% | 13% | 50% | 10% | [11,17,18]* |
| Uptake of self-tests among otherwise untested (to achieve 62% increase in testing) | 31% | 7.8% | 31% | 6.2% | [20] |
| Inter-reader agreement of HCVST | 97% | 98% | 88% | 97% | [8] |
| Receive NAT test after facility-based serologic test (assume reflex testing in Kenya and Vietnam) | 100% | 81% | 100% | 90% | [18,19] |
| Link to Care if NAT positive | 92% | 90% | 89% | 90% | [14,17]* |
| Start Treatment if linked to care | 92% | 90% | 96% | 90% | [14,17]* |
| Cured if start treatment | 95% | 74% | 92% | 98% | [17–19]* |
| Not tested for SVR | 0% | 25% | 5% | 0% | [17–19]* |
| *Cost parameters (2019 USD)* | | | | | |
| Cost of undertaking self-test (excluding test kit) | 15.46 | 3.00 | 10.00 | 2.52 | [11,21] |
| Facility-based RDT cost | 21.43 | 2.79 | 2.22 | 0.87 | |
| NAT test cost (diagnosis or SVR12) | 103.56 | 27.82 | 26.00 | 20.26 | |
| Pre-treatment costs–blood tests, liver disease staging, etc. [For Vietnam and China calculate as 10% of total treatment costs] | 123.29 | 40.89 | 171.40 | 157.00 | [19,22] |
| Average treatment cost | 1501.49 | 784.37 | 1542.60 | 1414.64 | [19,22,23] |

Parameters which do not vary by country are presented in Table C in S1 Text Sources marked with * represent programme data re-analysed for this study.

[20]. Therefore, the number of people undertaking self-tests is calculated as a function of the number accessing standard of care testing in each setting and assuming 10% of people that otherwise would access standard of care testing, use self-tests instead (substitution) as in HIVST models [26]. In addition, following linkage to care, we assume no difference in treatment initiation or success parameters between the standard of care vs self-testing scenarios, as a systematic review of HIV self-testing trials showed no difference in treatment initiation for those who were self-tested (risk ratio 0.98, 95% CI 0.86–1.11) [20]. Setting-based parameter assumptions, such as the standard of care costs, HCV prevalence, and cascades of care, are presented in Table 1.

**Sensitivity analysis.** Base case assumptions were varied in one-way sensitivity analysis, to reflect uncertainty in parameters and model structure, details are presented in Table 2. As in the base case, in each scenario explored in sensitivity analysis, we compared the introduction of HCVST to a counterfactual standard of care scenario in which no self-testing occurs.

## Costing

We gathered previously published costing data including from research studies and programme reports, identified through literature searches, previous research by the authors in each setting, and consultation with WHO focal points in each country (Table 1). Costs were identified from previous studies for HCV testing and treatment, and/or HIV self-testing in each setting from the healthcare providers' perspective. We aimed to identify costs which accounted for overheads, staff time, training, outreach, facilities, and start-up costs where available, however, costing methodology may vary by source. In two settings (Kenya and

**Table 2. Scenarios explored in one-way sensitivity analysis.** See main text for full details of base case assumptions.

| Scenario name | Description | Base case assumption |
|---|---|---|
| Direct to NAT | Patients are tested using NAT testing following a reactive self-test, without facility-based serology testing. Patients with an unclear or unknown result still receive a facility-based serological test. | Repeat serologic testing at facility |
| EIA standard of care | Standard of care antibody testing (including in the no self-testing counterfactual for this scenario) and repeat serologic testing are by enzyme immunoassays (EIA), which are more expensive than RDT (assumed double RDT test cost in Kenya and Vietnam, $32.76 in Georgia, and $5.07 in China [17,19,27]) and have sensitivity and specificity of 100% as they are the gold standard against which the RDTs are compared. | Standard of care antibody testing by RDT (95% sensitivity and 100% specificity) |
| Blood-based HCVST | Using a blood-based self-test based on the PMC First Response HCV Card Test (Premier Medical Corporation, Mumbai, India), which has an overall sensitivity of 96% (94–97%) and specificity of 99% (99–100%) [24]. The professional use version of this tests costs an average of $0.90 in the Global Fund pricing list [28]. We assume the self-test cost will be double this cost to account for additional costs of packaging, plus 25% overheads ($2.25) | HCVST by oral-fluid based test with 98% sensitivity and 100% specificity and costs $5.63 |
| High cost blood-based HCVST | Use blood-based self-test but assume the total cost is doubled to $4.50 due to uncertainty in the market price of the HCVST in each setting. | |
| High cost oral fluid HCVST | Double the cost of oral fluid-based HCVST to $11.25 | Oral fluid-based HCVST is $5.63 |
| Equal cost oral fluid HCVST | Set the cost of HCVST including distribution costs to be equal to the standard of care RDT test cost in each setting. | Distribution costs vary by setting (Table 1) |
| Low HCVST performance | Reduced HCVST performance to 90% sensitivity and 97% specificity, reflecting reduced test performance observed in samples from people co-infected with HIV [24] | HCVST 98% sensitivity and 100% specificity |
| High inter-reader agreement | Increase the inter-reader agreement to be 100% to reflect maximum successful usage of the self-tests | Inter-reader agreement varies by setting (88–98%) |
| High or low linkage | Assume 50% or 80% of positive self-tests link to facility-based repeat serological testing. Higher linkage to care is possible, particularly in Vietnam and Kenya, where testing is assumed to be peer-led. | 65% of positive self-tests link to facility |
| High or low HCVST uptake | Increase the uptake of self-testing to reach an 80% increase in overall testing or reduce the uptake of self-testing to reach only a 30% increase in overall testing | 62% increase in overall testing |
| High or low substitution | Vary the proportion of those using self-testing instead of facility-based testing to be 20% or 5% while keeping the increase in overall testing at 62%. | 10% substitution |
| Low or high self-test success | Vary the proportion of invalid self-test results to be 5% or 1% | 3% of self-test results invalid |

China), HCVST costs were assumed to be incremental adding to existing HIV self-testing programmes (Table 1).

Most cost data were identified in United States Dollars (USD) from between 2017–2019, with these being adjusted for inflation as necessary to present all costs in 2019 USD by using the consumer price index (CPI) for the study country [29]. The CPI value for Kenya was not available for 2019, so it was assumed to grow in the same ratio from 2018 as seen from 2017 to

2018. Some cost data from China were received in Chinese Yuan (RMB) [Chen, personal communication; Ong, personal communication], these were assumed to represent prices in 2019 and were converted to USD using the 2019 average exchange rate per USD (6.91 RMB per USD) from the International Monetary Fund's International Financial Statistics.

### Ethics statement

This study did not collect any primary data, it is based on data from previous studies and did not include any individual-level data or personal data, so did not require approval by an ethical review board.

## Results

### Costs and outcomes of the standard of care HCV testing

Without the HCVST intervention, the cost per HCV diagnosis (excluding treatment-related costs) is estimated to be $35 in Viet Nam, $55 in Georgia, $162 in China, and $361 in Kenya. The cost per cure is more comparable across settings—$1,238 in Georgia, $1,839 in China, $1,943 in Viet Nam, and $2,284 in Kenya, as the cost of treatment is similar across most settings (between $1,415–1,543 in Kenya, Viet Nam, and China, but approximately half at $784, in Georgia). Of note, there is a marked difference in the cost per person receiving facility-based testing, varying 25-fold from $0.87 in China to $21.43 in Kenya (Table 1), due to differences in test type and variation in consumable costs.

The absolute numbers of people diagnosed (Table 3) or cured (Table 4) in one year in the absence of HCVST are dependent on the population size and prevalence in each setting. In China, this is equivalent to 53 diagnosed and 41 cured per 100,000 MSM (antibody prevalence 1.0%); in Viet Nam 18,440 diagnosed and 14,440 cured per 100,000 PWID (antibody prevalence 66.0%); in Georgia 1,333 diagnosed and 801 cured per 100,000 men aged 40–49 (antibody prevalence 22.7%); and in Kenya 3,353 diagnosed and 2,691 cured per 100,000 PWID (antibody prevalence 12.9%).

### Cost and outcomes of HCVST in the base case

Tables 3 and 4 show the ICERs for the base case implementation of HCVST compared to the counterfactual of no HCVST. In addition to increasing the number of individuals diagnosed, introducing HCVST increases the cost per diagnosis in all settings. In the base case, it is assumed that introducing HCVST will increase the number of individuals tested by 62%, which increases the numbers diagnosed and cured by 30.6% in Viet Nam, 34.6% in Kenya,

**Table 3. Incremental cost per diagnosis of implementing HCVST (base-case analysis) in 2019 US dollars.** Note cost per diagnosis excludes treatment costs. The incremental cost per diagnosis for the HCVST scenario compared to no HCVST is presented for each of the four settings.

| Setting | Scenario | Total cost | Total diagnosed | Incremental cost | Incremental diagnosed | Incremental cost per diagnosis (ICER) |
|---------|----------|-----------|-----------------|------------------|----------------------|---------------------------------------|
| Kenya | No HCVST | $162,685 | 451 | - | - | - |
|  | Base case HCVST | $254,194 | 607 | $91,509 | 156 | $587 |
| Georgia | No HCVST | $171,037 | 3,123 | - | - | - |
|  | Base case HCVST | $349,430 | 4,216 | $178,393 | 1,093 | $163 |
| Viet Nam | No HCVST | $32,413 | 922 | - | - | - |
|  | Base case HCVST | $61,566 | 1,203 | $29,152 | 282 | $104 |
| China | No HCVST | $1,416 | 9 | - | - | - |
|  | Base case HCVST | $9,393 | 12 | $7,977 | 3.0 | $2,647 |

**Table 4. Incremental cost per cure, including cost of treatment, of implementing HCVST (base-case analysis) in 2019 US dollars.** The incremental cost per cure for the HCVST scenario compared to no HCVST is presented for each of the four settings.

| Setting | Scenario | Total cost | Total cured | Incremental cost | Incremental cured | Incremental cost per cure (ICER) |
|---|---|---|---|---|---|---|
| Kenya | No HCVST | $826,740 | 362 | - | - | - |
| | Base case HCVST | $1,147,729 | 487 | $320,989 | 125 | $2,566 |
| Georgia | No HCVST | $2,322,642 | 1,876 | - | - | - |
| | Base case HCVST | $3,254,115 | 2,533 | $931,473 | 657 | $1,418 |
| Viet Nam | No HCVST | $1,402,615 | 722 | - | - | - |
| | Base case HCVST | $1,850,412 | 943 | $447,797 | 221 | $2,030 |
| China | No HCVST | $12,785 | 7 | - | - | - |
| | Base case HCVST | $26,690 | 9 | $11,905 | 2.4 | $4,956 |

35.0% in Georgia, and 34.6% in China (Fig 2), due to slight differences in the cascade of care in each setting (see Table 1).

The ICER per additional person diagnosed with the introduction of HCVST is lowest in Viet Nam ($104), $163 in Georgia, $587 in Kenya, and $2,647 in China (Table 3). The variations in the cost per diagnosis by setting relate to the differences in HCV prevalence in each study setting, with cheaper costs in the settings with higher prevalence. The ICER per person cured ranges from $1,418 in Georgia, to $2,030 in Viet Nam, $2,566 in Kenya, and $4,956 in China (Table 4). The HCVST cost per cure is driven by treatment costs in each setting.

## Sensitivity analysis

The differences in the ICER per diagnosis and cure for HCVST under the sensitivity analysis scenarios are shown in Fig 3 and Figs A-C in S1 Text, compared to the base case. Similar patterns are seen across all settings.

The cost per diagnosis (viraemic infection) (Fig 3) is highly sensitive to the cost of HCVST. The largest decrease in the ICER per diagnosis is seen when the HCVST price is matched to the standard of care RDT test cost, except for Kenya where this increases the ICER slightly, due to the high standard of care test cost. The ICER per diagnosis increases the most when the cost of the HCVST increases, there is low uptake of HCVST, when linkage is low, there is low performance of the HCVST or high substitution of standard of care tests with self-tests. This pattern holds across all countries, with the EIA as standard of care scenario also increasing the ICER per diagnosis in Georgia and Kenya. Low self-test success has little impact on the ICER. Reductions are seen in the ICER with use of the blood-based HCVST, even at double its usual price, as well as when there is high uptake of HCVST, high linkage to facility-based testing, high inter-reader agreement, and low substitution of standard of care tests with self-tests. Proceeding from a positive HCVST direct to NAT testing also reduces the ICER in all settings.

Differences in the ICERs in sensitivity analysis are driven by changes in the number of individuals diagnosed and the total cost. The number diagnosed is influenced most by changes in the uptake of HCVST and linkage to healthcare facilities after self-testing.

The sensitivity of the ICER per cure to changes in parameters within each setting is impacted by the same factors as the cost per diagnosis. Although the relative magnitudes differ slightly, the pattern is the same as seen in the cost per diagnosis (Fig C in S1 Text).

## Discussion

We evaluated the cost-effectiveness of HCVST compared to facility-based testing across four settings and populations with a wide range of HCVAb prevalence. Based on the assumptions

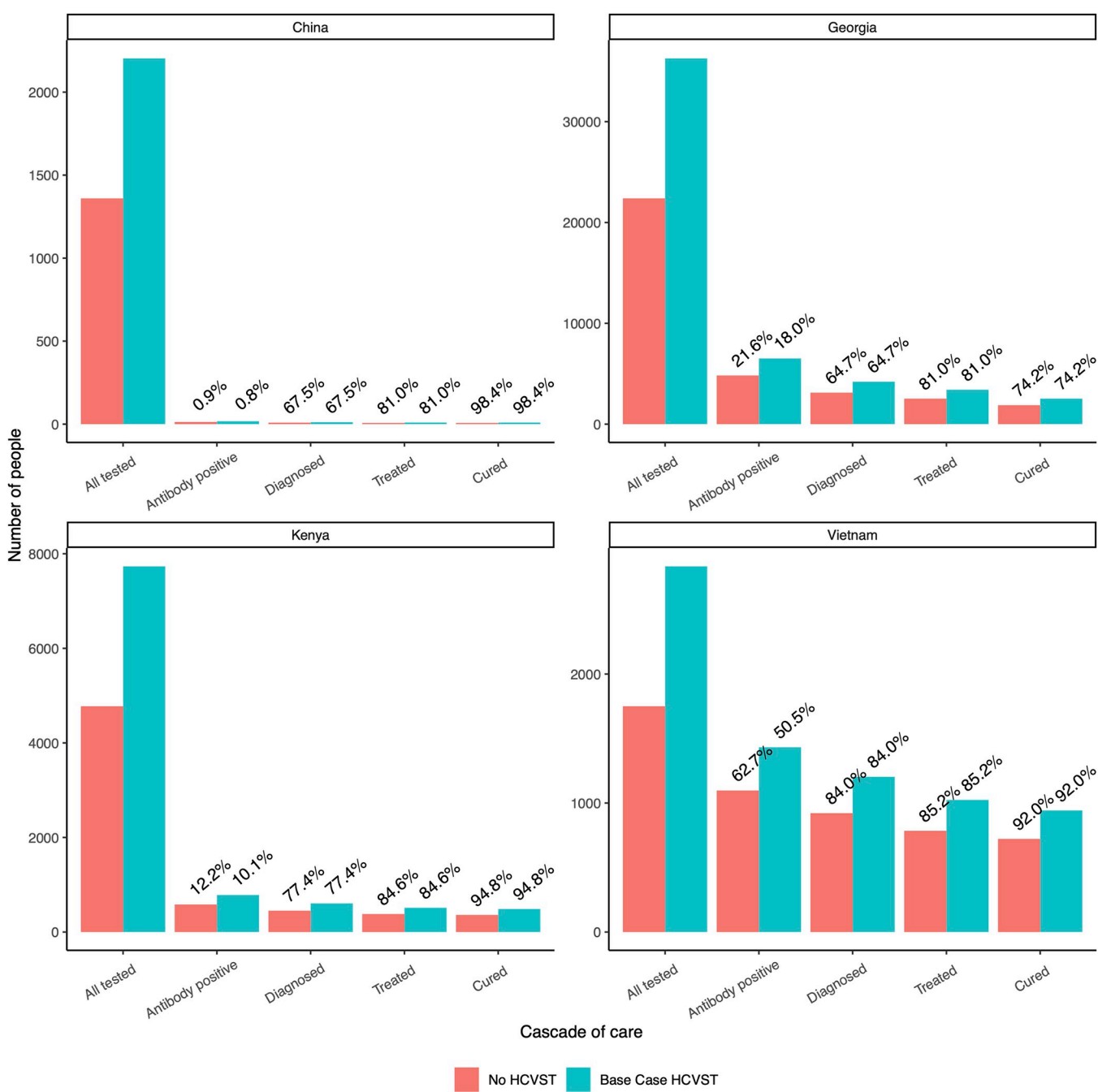

**Fig 2. Cascade of care of patients tested, antibody positive, diagnosed viraemic, treated, and cured in each setting for the standard of care with no HCVST compared to the introduction of HCVST (base case analysis).** Values above each bar show the percent of previous step in cascade of care within standard of care or base case cascades (eg. percent antibody positive out of all tested, percent diagnosed viraemic out of antibody positive).

that introduction of HCVST can increase the number of people who know their status, are diagnosed with chronic HCV, and are successfully treated, we found that incremental cost per HCV diagnosis for adding HCVST to the facility-based testing standard of care varied widely by setting, from $104 in Viet Nam to $2,647 in China, while cost per cure was lowest in Georgia ($1,418), and highest in China ($4,956). In all settings, HCVST resulted in more people

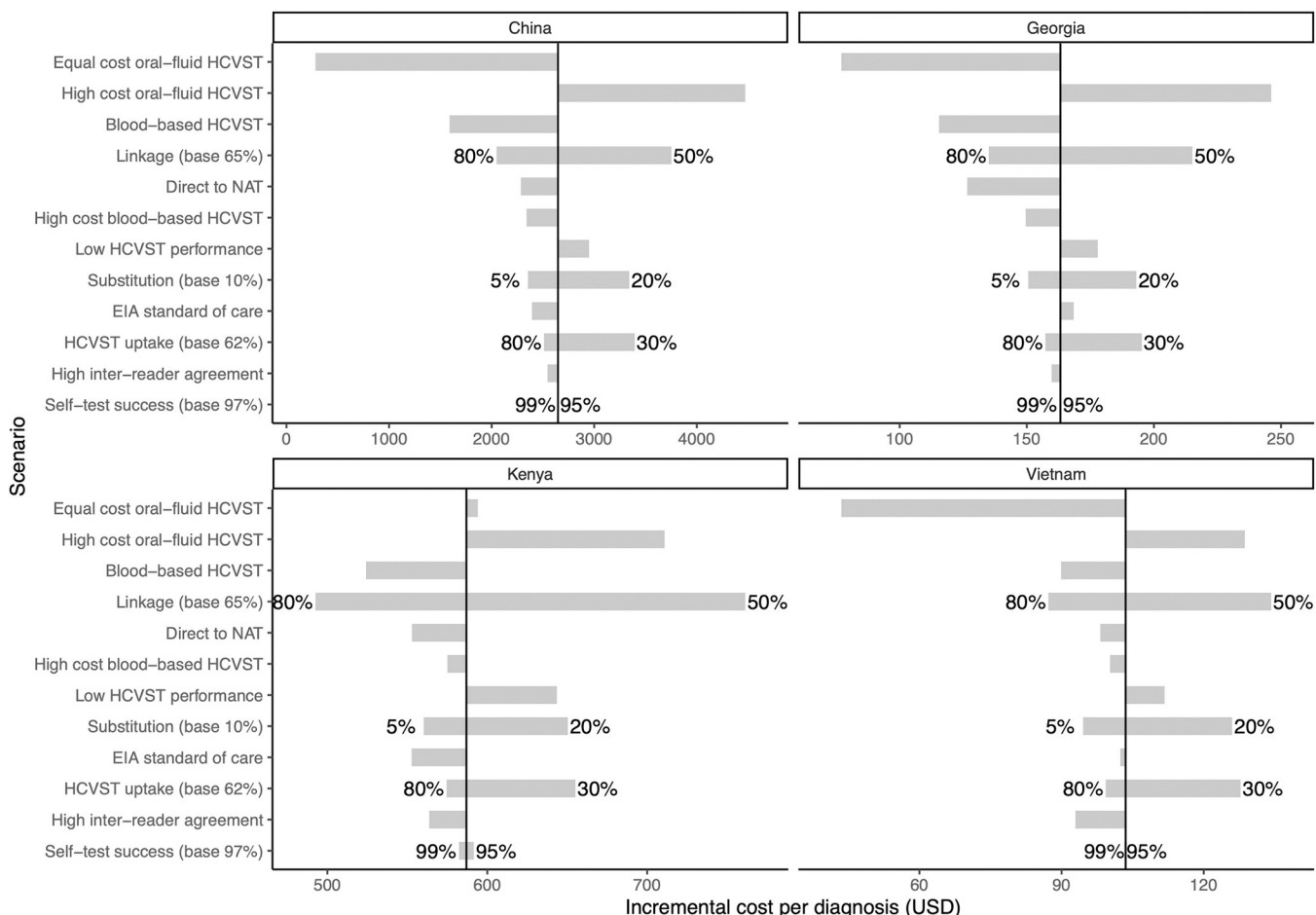

**Fig 3. Tornado plot showing sensitivity analyses of how changes in parameter assumptions effects the incremental cost per diagnosis for each country.**
The sensitivity analyses are described in detail in Table 2. Note that the x-axis scale is different for each country, but order of y-axis is the same. * The vertical line represents the base case incremental cost per diagnosis, as shown in Table 3. The end of each bar represents the incremental cost per diagnosis in each modelled scenario, with the length of the bar representing the magnitude of the difference from the base case. Numbers at end of bars are the alternative values for those parameters in that sensitivity analysis.

diagnosed or cured compared to the standard of care, at a higher cost, meaning it is not cost-saving. Variation in cost per diagnosis between settings is due to differences in prevalence and test costs, while differences in the cost per cure were driven primarily by treatment costs rather than the cost of diagnosis.

The ICER (cost per diagnosis or cost per cure) of HCVST is impacted strongly by the price of the HCVST itself, and the uptake of the tests, with higher uptake leading to a reduction in the ICER. Conversely, with greater substitution of standard of care tests by self-tests, the ICER increases due to the higher cost of HCVST compared to facility-based RDT testing. The tests' performance and usability in terms of inter-reader agreement had little impact on the ICER due to generally high values for these parameters. If we do not undertake confirmatory HCV antibody testing at the facility but instead go direct to NAT, then the ICER decreases slightly because of decreases in test costs and increases in the number of people diagnosed from avoiding some false negatives resulting from the facility-based RDT.

This study is the first to evaluate the potential cost and impact of HCVST in terms of increasing access to HCV diagnosis and cure, and so by necessity, we had to make assumptions

about some parameters. However, our study is strengthened because we used real-world examples for the case study settings, using locally observed costs, HCV prevalence, HCVST feasibility trials, and cascades of care for the standard of care pathway. Where local estimates were not available for undertaking HCVST, we drew on work for HIVST in the same populations, including adapting the costs of implementation of HIVST in Kenya and China.

This study has several limitations, particularly in uncertainty around parameters regarding uptake of self-testing, linkage to confirmatory testing, and in the lack of information about how HCVST will be implemented. As this study focused on the potential implementation of HCVST in different settings under different structural assumptions, we focused on one-way sensitivity analysis, and did not include a probabilistic analysis of the joint effect of parameters on the cost-effectiveness. We used data from HIV self-testing where possible, but differences between the diseases could affect the relevance of this data for HCV self-testing. In addition, the four specific case studies may not be broadly generalisable. However, they were selected to represent different populations (PWID, MSM, general population), a wide range of HCV prevalence, and different testing regimes. The current availability of testing and treatment for HCV in each setting will also affect the observed cascades of care identified, and these rates are likely to change over time as each setting progresses towards elimination. Our analysis assumes that there is capacity within the system for scaling up diagnosis and treatment, which ranges from an additional 3 diagnosed patients in the China setting to 1,093 additional patients in the Georgia setting in the base case, and that costs will scale linearly with the number of patients treated. In addition, the study only uses a one-year time horizon, focusing on the outcomes of the number of people diagnosed and cured. We do not capture the long-term benefits of diagnosis and curing people of HCV, which will lead to reduced morbidity and mortality from end-stage liver disease and reduce onward transmission, as well as avert costs of liver disease care. In this preliminary study, it was not feasible to incorporate a model of disease progression and HCV transmission.

Although there is currently limited data and experience of using HCVST, there are key transferable lessons from work on HIV. For example, with both HIV and HCV there may be an impact of stigma on uptake of testing, multi-step testing is required, both HIV and HCV affect similar key populations, and a significant proportion maybe co-infected. However, at present there is a much higher proportion of people living with HCV who do not know their status (80%) compared to around 20% for HIV. Self-testing is useful to target those who are unlikely to otherwise access care, and this may become increasingly important as countries get closer to reaching HCV elimination targets. The possibility of a cure for HCV compared to lifelong treatment for HIV, and the largely asymptomatic nature of HCV infection will also impact testing uptake.

In the early days of HIVST research, modelling studies predicted that HIVST would be cost-effective [26] and subsequent studies confirmed this. Cost-effectiveness studies of HIVST in LMIC also found the introduction of self-testing led to increased diagnosis and linkage to care, but with an additional cost to identify each case [21,26]. The first HCVST real world implementation studies are currently underway, piloting different models of HCVST distribution in Georgia, Malaysia and Pakistan [30–32]. Costing and cost-effectiveness analysis based on these and future studies will allow policymakers to make an informed decision on optimal approaches to implement HCVST in each setting. Future work on HCVST should aim to present outcomes in terms of cost per quality adjusted life year or disability adjusted life year to allow decision-makers to compare value for money across different types of interventions. Our results indicate that the introduction of HCVST may increase the overall numbers of HCV-infected people diagnosed and cured, but will require additional investment compared to the current standard of care facility-based testing and treatment pathway. These additional costs

need to be minimised through ensuring that test costs are kept low and linkage to care rates are high.

## Supporting information

**S1 Checklist. This file is a CHEERS checklist.**
(PDF)

**S1 Text. This file contains supplementary methods outlining case study assumptions and current testing guidelines in each setting, three supplementary figures (Figs A-C), and three supplementary tables (Tables A-C).**
(DOCX)

## Author Contributions

**Conceptualization:** Josephine G. Walker, Muhammad S. Jamil, Emmanuel Fajardo, Niklas Luhmann, Sonjelle Shilton.

**Data curation:** Josephine G. Walker, Elena Ivanova, Muhammad S. Jamil, Jason J. Ong, Fern Terris-Prestholt.

**Formal analysis:** Josephine G. Walker.

**Funding acquisition:** Elena Ivanova, Sonjelle Shilton.

**Investigation:** Josephine G. Walker.

**Methodology:** Jason J. Ong, Cheryl Case Johnson, Fern Terris-Prestholt, Peter Vickerman.

**Project administration:** Josephine G. Walker.

**Resources:** Elena Ivanova, Sonjelle Shilton.

**Software:** Josephine G. Walker.

**Supervision:** Muhammad S. Jamil, Philippa Easterbrook, Niklas Luhmann, Peter Vickerman, Sonjelle Shilton.

**Visualization:** Josephine G. Walker.

**Writing – original draft:** Josephine G. Walker.

**Writing – review & editing:** Muhammad S. Jamil, Jason J. Ong, Philippa Easterbrook, Emmanuel Fajardo, Cheryl Case Johnson, Niklas Luhmann, Fern Terris-Prestholt, Peter Vickerman, Sonjelle Shilton.

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
