## [Decision Letter · Decision Letter 0]

17 Oct 2022

PGPH-D-22-01373

Cost and cost-effectiveness of Hepatitis C virus self-testing in four settings: an economic evaluation

Dear Dr. Walker,

Thank you for submitting your manuscript to PLOS Global Public Health. After careful consideration, we feel that it has merit but does not fully meet PLOS Global Public Health’s publication criteria as it currently stands. Therefore, we invite you to submit a revised version of the manuscript that addresses the points raised during the review process.

We look forward to receiving your revised manuscript.

Kind regards,

Alice Zwerling, PhD

Academic Editor

Journal Requirements:

2. Please provide separate figure files in .tif or .eps format only and remove any figures embedded in your manuscript file. Please also ensure that all files are under our size limit of 10MB.

3. We do not publish any copyright or trademark symbols that usually accompany proprietary names, eg ©, ®, or ™  (e.g. next to drug or reagent names). Please remove all instances of trademark/copyright symbols throughout the text, including ® on page 9.

4. We have noticed that you have uploaded Supporting Information files, but you have not included a list of legends. Please add a full list of legends for your Supporting Information files after the references list. 

Additional Editor Comments (if provided):

Reviewers' comments:

Reviewer's Responses to Questions

**Comments to the Author**

1. Does this manuscript meet PLOS Global Public Health’s publication criteria? Is the manuscript technically sound, and do the data support the conclusions? The manuscript must describe methodologically and ethically rigorous research with conclusions that are appropriately drawn based on the data presented.

Reviewer #1: Yes

Reviewer #2: Yes

Reviewer #3: Partly

2. Has the statistical analysis been performed appropriately and rigorously?

Reviewer #1: Yes

Reviewer #2: Yes

Reviewer #3: I don't know

3. Have the authors made all data underlying the findings in their manuscript fully available (please refer to the Data Availability Statement at the start of the manuscript PDF file)?

Reviewer #1: Yes

Reviewer #2: Yes

Reviewer #3: Yes

4. Is the manuscript presented in an intelligible fashion and written in standard English?

Reviewer #1: Yes

Reviewer #2: Yes

Reviewer #3: Yes

5. Review Comments to the Author

Reviewer #1: The authors incorporated most of the changes requested in the previous review, and the article meets all the technical requirements to be published. However, the originality of the work in the field of knowledge is low.

Reviewer #2: This paper looks at the cost and cost-effectiveness of an emerging technology to increase diagnosis and treatment uptake for hepatitis C. The study is timely given the need to engage new populations in hepatitis C care, and will be useful to guide decisions on how to test people not engaged in care. The methods are suitable for the study aims, appropriate sensitivity analyses are conducted and the manuscript is clear and well written. While there are some assumptions, they are reasonably justified based on HIV and other experiences and necessary in lieu of pilots. I have some minor points below.

1. There is not a lot of information provided on the cost estimates. E.g. it is noted that they are for HCV testing and treatment, and include overheads, staff time, training, outreach, facilities, and start-up costs. Some further details could be provided on the breakdown of costs, rather than single totals.

2. The outcome measures are based on cost per diagnosis and cost per treatment, which is not a problem but can be difficult to benchmark. Perhaps in Table 3 it would be useful to report the baseline cost per diagnosis and cost per treatment (i.e. without self-testing)? This information can be obtained from the table, so it is just being more explicit.

3. In Table 3, are the numbers diagnosed and treated in each year influenced by the stage of elimination that these countries are up to? In the text it is noted that these are influenced by population size and prevalence, but also depending on how long DAAs have been available these numbers may be abnormally high or may have stabilized.

Reviewer #3: Reviewer comments to manuscript

"Cost and cost-effectiveness of Hepatitis C virus self-testing in four settings: an economic evaluation"

The title is repetitive it should not include the words “… an Economic evaluation” and “Cost” since a cost effectiveness is an economic evaluation study. In addition, it is implicit that a cost effectiveness study includes an estimate and analysis of costs.

On Figure 1: The decision tree requires to be reported as in conventional notation showing decision and probabilistic nodes. In addition, highlight the probabilistic values assumed in estimating the expected value of number of diagnosed and cure cases as well as the costs. In the alternative of no tested in the decision tree it is not clear whether this is a relevant alternative to be evaluated since the relevant comparator of the value of HCVST is not to use it. In addition, if the relevant outcome is to be aware of the HCV status that means that being positive or negative is not the relevant outcome but to be aware of its health status. Seems to be confusing as it is reported in the current manuscript, see decision tree from reference of Schackman BR 2015

Schackman BR, Leff JA, Barter DM, DiLorenzo MA, Feaster DJ, Metsch LR, Freedberg KA, Linas BP. Cost-effectiveness of rapid hepatitis C virus (HCV) testing and simultaneous rapid HCV and HIV testing in substance abuse treatment programs. Addiction. 2015 Jan;110(1):129-43. doi: 10.1111/add.12754. PMID: 25291977; PMCID: PMC4270906.

Tables 3 and 4 should have some explaining nots to interpret results reported.

Table 3 specify these are US dollars 2019

Table 4 is supposed to include treatment costs modify title as corresponds

6. PLOS authors have the option to publish the peer review history of their article (what does this mean?). If published, this will include your full peer review and any attached files.

**Do you want your identity to be public for this peer review?** For information about this choice, including consent withdrawal, please see our Privacy Policy.

Reviewer #1: No

Reviewer #2: No

Reviewer #3: No

---

## [Decision Letter · Decision Letter 1]

19 Dec 2022

PGPH-D-22-01373R1

Cost-effectiveness of Hepatitis C virus self-testing in four settings

Dear Dr. Walker,

Thank you for submitting your manuscript to PLOS Global Public Health. After careful consideration, we feel that it has merit but does not fully meet PLOS Global Public Health’s publication criteria as it currently stands. Therefore, we invite you to submit a revised version of the manuscript that addresses the points raised during the review process.

We look forward to receiving your revised manuscript.

Kind regards,

Alice Zwerling, PhD

Academic Editor

Journal Requirements:

Additional Editor Comments (if provided):

Reviewers' comments:

Reviewer's Responses to Questions

**Comments to the Author**

1. If the authors have adequately addressed your comments raised in a previous round of review and you feel that this manuscript is now acceptable for publication, you may indicate that here to bypass the “Comments to the Author” section, enter your conflict of interest statement in the “Confidential to Editor” section, and submit your "Accept" recommendation.

Reviewer #1: All comments have been addressed

Reviewer #2: All comments have been addressed

Reviewer #3: All comments have been addressed

2. Does this manuscript meet PLOS Global Public Health’s publication criteria? Is the manuscript technically sound, and do the data support the conclusions? The manuscript must describe methodologically and ethically rigorous research with conclusions that are appropriately drawn based on the data presented.

Reviewer #1: Yes

Reviewer #2: Yes

Reviewer #3: Yes

3. Has the statistical analysis been performed appropriately and rigorously?

Reviewer #1: Yes

Reviewer #2: Yes

Reviewer #3: No

4. Have the authors made all data underlying the findings in their manuscript fully available (please refer to the Data Availability Statement at the start of the manuscript PDF file)?

Reviewer #1: Yes

Reviewer #2: Yes

Reviewer #3: Yes

5. Is the manuscript presented in an intelligible fashion and written in standard English?

Reviewer #1: Yes

Reviewer #2: Yes

Reviewer #3: Yes

6. Review Comments to the Author

Reviewer #1: Several problems with the model are observed. First, the authors assume that the four scenarios have the possibility of increasing care in medical units to confirm cases with the development of tests such as NAT. In addition, they consider that it is possible to provide medical care and treatment to the population diagnosed with self-diagnostic tests (human resources, infrastructure, and budget). Adopting more realistic or conservative coverage levels for each locality where the cost-effectiveness model is being carried out is suggested.

Among the most robust assumptions that would have to be assumed is that the chain/sequence of care can be fulfilled by the health services that are being proposed.

Another situation to consider is the percentage of acceptance of the test and if people intend to take it to find out their condition.

It is necessary to validate several of the parameters of the China scenario because it is the location with the highest cost. Also, find out if these costs are due to low availability or other reasons.

The modifications in the efficacy of the self-application tests are minimal, which can determine the non-variability or impact on the ICER. It is suggested to incorporate variations of effectiveness in actual care conditions for this type of test.

Indeed, the authors did not calculate the future impact, but they did not calculate the probability of getting sick again. Additionally, due to the type of disease and population, it would be necessary to reapply for these tests annually.

Supplement 1

It is recommended to search for values published in the literature of other programs rather than applying their assumptions to these parameters.

Reviewer #2: I have no further comments.

Reviewer #3: 2nd review round of the manuscript

Cost-effectiveness of Hepatitis C virus self-testing in four settings

General comments

1. It seems that figure 1 explanation in lines 143-153 have to be separated from title of the figure 1.

2. In page 11 (lines 159) authors refer to figure 1A but if they replace the previous figure 1A and 1B with the new Figure 1 of the decision tree which is only one. Make corrections as correspond.

Methodology

1. The probabilities of the decision tree are described in table 1 and supplementary tables 1-3. In the case of supplementary tables 1 and 2 the values are not reported because they vary by country?

Results and SA

1. Authors say in base case that with introduction of HCVST the number of individuals diagnosed will increase by 62%, is that a target or is it estimated from table 1?

2. There is a last comment on the sensitivity analysis. The lack of a probabilistic sensitivity when all parameters are varied for an estimate of the joint effect on cost effectiveness should be reported as a limitation of the study. Otherwise this analysis should be conducted.

Discussion (Interpretation)

1. Authors say that in base case scenario the arm for self-testing includes 4 alternatives. For base case estimates this is not a problem: the model begins with self-test then positives are confirmed with serological test then positives with NAT. But this is not clear for sensitivity analysis. In SA the starting test is self-test then those positive go direct to NAT? The branch for unclear/unknown which go direct also to NAT? And then the idea is that additional cases are diagnosed from this group of unclear/unknown, something which the alternative of facility based test does not consider, in addition to other reasons?

7. PLOS authors have the option to publish the peer review history of their article (what does this mean?). If published, this will include your full peer review and any attached files.

**Do you want your identity to be public for this peer review?** For information about this choice, including consent withdrawal, please see our Privacy Policy.

Reviewer #1: No

Reviewer #2: No

Reviewer #3: No

---

## [Decision Letter · Decision Letter 2]

22 Feb 2023

Cost-effectiveness of Hepatitis C virus self-testing in four settings

PGPH-D-22-01373R2

Dear Dr Walker,

We are pleased to inform you that your manuscript 'Cost-effectiveness of Hepatitis C virus self-testing in four settings' has been provisionally accepted for publication in PLOS Global Public Health.

Best regards,

Alice Zwerling, PhD

Academic Editor

Reviewer Comments (if any, and for reference):

Reviewer's Responses to Questions

**Comments to the Author**

1. If the authors have adequately addressed your comments raised in a previous round of review and you feel that this manuscript is now acceptable for publication, you may indicate that here to bypass the “Comments to the Author” section, enter your conflict of interest statement in the “Confidential to Editor” section, and submit your "Accept" recommendation.

Reviewer #1: All comments have been addressed

Reviewer #2: All comments have been addressed

Reviewer #3: All comments have been addressed

2. Does this manuscript meet PLOS Global Public Health’s publication criteria? Is the manuscript technically sound, and do the data support the conclusions? The manuscript must describe methodologically and ethically rigorous research with conclusions that are appropriately drawn based on the data presented.

Reviewer #1: Yes

Reviewer #2: Yes

Reviewer #3: Yes

3. Has the statistical analysis been performed appropriately and rigorously?

Reviewer #1: Yes

Reviewer #2: Yes

Reviewer #3: Yes

4. Have the authors made all data underlying the findings in their manuscript fully available (please refer to the Data Availability Statement at the start of the manuscript PDF file)?

Reviewer #1: Yes

Reviewer #2: Yes

Reviewer #3: Yes

5. Is the manuscript presented in an intelligible fashion and written in standard English?

Reviewer #1: Yes

Reviewer #2: Yes

Reviewer #3: Yes

6. Review Comments to the Author

Reviewer #1: the new version of the paper incorporated the recommendations made previously. In addition to the fact that the work is interesting and relevant, I recommend its publication

Reviewer #2: The authors have addressed my comments and I have nothing further to add. I think this paper makes a nice contribution to the literature.

Reviewer #3: All the comments I made in last second round were addressed and consider that the manuscript is ready for publication.

7. PLOS authors have the option to publish the peer review history of their article (what does this mean?). If published, this will include your full peer review and any attached files.

**Do you want your identity to be public for this peer review?** For information about this choice, including consent withdrawal, please see our Privacy Policy.

Reviewer #1: No

Reviewer #2: No

Reviewer #3: No
